# ADOPD-INSTRUCT: A LARGE-SCALE MULTIMODAL DATASET FOR DOCUMENT EDITING

**[Masking - Text Element]**

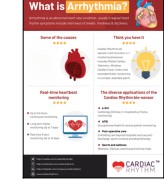

**Instr**: Locate the "Some of the causes" section. Delete the text with red bullets in this section including the red bullets.

**[Masking - Non-Text Element]**

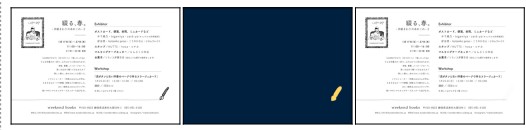

**Instr**: Locate the pen illustration at the bottom right of the document, near the line above the footer. Remove the pen illustration entirely.

**[Inpainting - Text Element]**

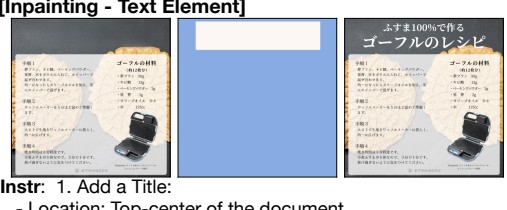

**Instr**: 1. Add a Title:
  - Location: Top-center of the document.
  - Content: "ふすま100%で作る\nゴーフルのレシピ".
2. Adjust Text Alignment:
  - Ensure the newly added title is centered at the top of the document, above the existing text.

**[Inpainting - Non-Text Element]**

**Instr**: Add a cartoon image of a boy with a big smile and two fingers up in a peace sign to the top right corner. Ensure the boy's head and arms are visible, and the cartoon figure should not overlap with the text in the top or in the middle.

Figure 1: We introduce ADOPD-Instruct, a large-scale multimodal dataset designed for document editing tasks. ADOPD-Instruct includes comprehensive instructions for entity-level editing, encompassing both textual content and non-text design elements in visually-rich documents. Each example includes the original document, the segmentation mask indicating the element to be edited, the target document after editing, and a human-curated instruction.

## ABSTRACT

Visually-rich document editing is a complex multimodal task with a wide range of real-world applications. Despite increasing interest, there is a significant lack of publicly available datasets offering detailed entity-level annotations and step-by-step instructions for the editing process. To address this, we introduce ADOPD-Instruct, a multimodal dataset designed specifically for document editing tasks. ADOPD-Instruct includes visually-rich documents, precise entity-level masks highlighting elements to be edited, and step-by-step edit instructions, targeting both the masking and inpainting processes for text and non-text design elements. ADOPD-Instruct instructions have been carefully curated by human annotators to ensure high quality across the dataset. We conduct extensive evaluations of current Multimodal Large Language Models (MLLMs) and image editing models using various image backbones to assess their performance on document editing. The results reveal substantial challenges: current MLLMs struggle to generate accurate and detailed instructions, while image editing models often fail to follow instructions precisely, particularly with text edits. These findings underscore the limitations of existing models and highlight the importance of annotated datasets like ADOPD-Instruct for advancing this domain. Dataset is available at: https://huggingface.co/datasets/adopd-instruct/ADOPD-Instruct.

# 1 INTRODUCTION

Visually-rich document editing is a crucial task with a wide range of downstream applications, from automated document generation to personalized document creation, where precise edits can significantly impact the quality and effectiveness of the final product. Despite its importance, progress in automated document editing has been limited, largely due to the lack of comprehensive document datasets with fine-grained, entity-level dense annotations of edits across different modalities.

Previous efforts in creating visually rich document datasets (Zhong et al., 2019; Li et al., 2020; Pfitzmann et al., 2022; Cheng et al., 2023) have focused primarily on annotating document labels for layout analysis. These annotations, which include categories like title, section, paragraph, figure, and table, are more suited for layout manipulations than for content editing. In contrast, ADOPD (Gu et al., 2024), a public document decomposition dataset, introduce entity-level annotations that better align with document editing tasks. However, ADOPD lacks accompanying annotations or instructions for the editing process, limiting its applicability. Recently, DocEdit (Mathur et al., 2023) offers a fixed set of commands for editing, but the usecase is retricted to structured document files.

To address the issue of data scarcity, we introduce ADOPD-Instruct, a publicly available multimodal dataset with detailed annotations and step-by-step instructions for entity-level editing in visually-rich documents. Table 1 compares other visually-rich document datasets with our ADOPD-Instruct. ADOPD-Instruct is built upon ADOPD (Gu et al., 2024) documents. We first use GPT-4o (OpenAI, 2024) to generate initial editing instructions, which are then refined by human annotators to ensure accuracy and validity. Given the complexity of document editing, ADOPD-Instruct focuses on two key document editing processes, namely *Masking* and *Inpainting*. Recognizing the multimodal nature of the task, we treat text and non-text element editing as distinct subtasks, collecting separate instructions for each.

Table 1: Comparison of ADOPD-Instruct with related document datasets.

| Dataset | Size | Segmentation? | Instr.? |
|---|---|---|---|
| PubLayNet | 360k | Layout-level | ✗ |
| DocBank | 500k | Layout-level | ✗ |
| DocLayNet | 81k | Layout-level | ✗ |
| M6Doc | 9k | Layout-level | ✗ |
| ADOPD | 120k | Entity-level | ✗ |
| DocEdit | 28k | Layout-level | ✓ |
| ADOPD-Instruct | 181k | Entity-level | ✓ |

We conduct extensive experiments to evaluate the performance of current models on visually-rich document understanding and editing tasks. Human assessments of GPT-4o-generated instructions indicate that while the model demonstrates considerable potential, there are still common errors such as inaccurate descriptions, incomplete edits, and omissions of crucial details for reconstruction. We further evaluate eight open-source multimodal large language models (MLLMs) on a simplified document editing setup where only a single text or non-text design element is edited. Experimental results show that the instructions generated by current open-source MLLMs did not fully achieve the level of detail and precision found in human-written instructions when describing intricate edits between visually-rich documents.

We further evaluate four image editing models on instruction-guided document editing tasks. The results indicate that these models face challenges in following detailed, multi-step instructions, partially due to the gap between the long and complex instructions required for document editing and the simpler, single-step instructions on which the models were originally trained. Additionally, we observe that Stable Diffusion-based models encountered difficulties when inpainting text elements, further highlighting the limitations of current models in handling document-specific edits. These findings emphasize the need for continued research and the development of datasets like ADOPD-Instruct, which can provide more suitable benchmarks for advancing document editing capabilities.

Our contributions are summarized as follows:

- We curate ADOPD-Instruct, a large-scale multimodal dataset with entity-level annotations and step-by-step instructions for visually-rich document editing, with a particular emphasis on the *Masking* and *Inpainting* of both text elements and non-text design components.
- We conduct extensive empirical studies on ADOPD-Instruct to evaluate the performance of state-of-the-art MLLMs in visually-rich document understanding, as well as the efficacy of leading image editing models in document editing tasks.
- We highlight the limitations of current MLLMs and image editing models in performing visually-rich document editing tasks, emphasizing the necessity for more sophisticated methodologies to enhance model performance in this domain.

## 2 RELATED WORK

### 2.1 VISUALLY-RICH DOCUMENT DATASET

Visually-rich document (VRD) datasets are essential for advancing document study. Pub-LayNet (Zhong et al., 2019), DocBank (Li et al., 2020), DocLayNet (Pfitzmann et al., 2022) and M$^6$Doc (Cheng et al., 2023) provide large-scale labeled datasets for understanding document layout structures, focusing on the segmentation of elements like paragraphs, images, and tables. RVL-CDIP (Harley et al., 2015) and FUNSD (Jaume et al., 2019) focus on document classification and form understanding, enabling models to handle complex documents with varied formats. XFUND (Xu et al., 2022) incorporates multilingual annotations and entity linking for visually complex forms. The ADOPD dataset (Gu et al., 2024) enhances document analysis with high-quality document images and dense annotations for visual entities and text bounding boxes. DocEdit (Mathur et al., 2023) explores document editing using a fixed set of commands, but focus more on modifying structured document files. Related to this line, TRIN (Zhang et al., 2024) collects a dataset with text-rich images with captions, text bounding boxes, and QA instructions; and LayoutLLM (Luo et al., 2024) specifically incorporates layout-aware information in its training dataset related to documents. While much efforts on dataset have been made in VRD, there is currently no publicly available dataset specifically tailored for fine-grained entity-level document editing or generation. This gap highlights the need for comprehensive datasets that facilitate diverse fine-grained editing tasks in visually-rich documents.

### 2.2 INSTRUCTION-GUIDED IMAGE EDITING

Instruction-guided image editing has gained significant attention due to its potential for intuitive, user-driven modifications. Early approaches, such as GAN-based models (Isola et al., 2017) and VAEs (Kingma & Welling, 2014), primarily target tasks like object removal, inpainting, and style transfer. Methods like DeepFill (Yu et al., 2018; 2019) and EdgeConnect (Nazeri et al., 2019) advance inpainting by utilizing contextual cues, though they typically require manual masks or simple prompts. LaMa (Suvorov et al., 2022), incorporating Fast Fourier Convolutions, further improved natural image inpainting and removal tasks. Recent advancements in instruction-based image editing leverage multimodal models such as Stable Diffusion (Rombach et al., 2022), InstructPix2Pix (Brooks et al., 2023), and DALLE-2 (Ramesh et al., 2022), and excel in scene-level manipulation. However, existing models are primarily trained on natural image datasets and struggle with fine-grained entity-level editing in visually-rich documents, particularly when handling text and complex design elements. Text-Diffuser (Chen et al., 2023a; 2024a) has made progress in text generation using diffusion models, but it faces challenges in generating or editing longer and more complex text sequences, which is a common scenario in document editing. GlyphDraw (Ma et al., 2023; 2024) investigates text rendering in image generation by conditioning on glyph information. DnD-Transformer (Chen et al., 2024b) introduces an innovative depth dimension for autoregression alongside the traditional spatial dimension, demonstrating potential for improving text rendering in image generation tasks. Overall, most existing models are optimized for natural images and lack the multimodal reasoning and fine-grained understanding required for editing intricate document structures that involve both text and design components. Various datasets have been proposed for image editing tasks, including MagicBrush (Zhang et al., 2023), Emu Edit (Sheynin et al., 2024), HQ-Edit (Hui et al., 2024), and UltraEdit (Zhao et al., 2024). However, the source images in these datasets primarily consist of natural images from databanks such as MSCOCO, or model-generated images. These sources differ significantly from the document images used in our dataset, highlighting the unique nature and focus of our work.

## 3 ADOPD-INSTRUCT DATASET

### 3.1 TASK FORMULATION FOR VISUALLY-RICH DOCUMENT EDITING

ADOPD-Instruct is a multimodal dataset curated for the intrinsic entity-level editing in visually-rich documents. We decompose the document editing process into two primary tasks: *Masking* and *Inpainting*. Figure 1 provides illustrative examples of these two tasks. Each data instance in ADOPD-Instruct comprises a visually-rich document image $\mathbf{I}_{\text{doc}}$, a detailed step-by-step instruc-

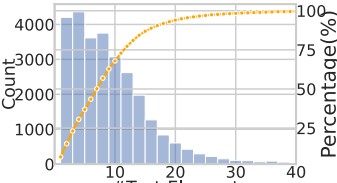 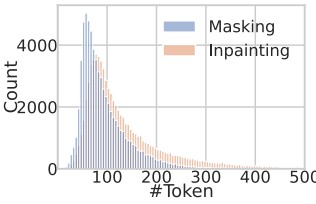 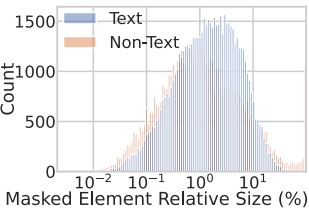

(a) Annotated #text-element for each document.

(b) #Token in instructions for the two document editing tasks.

(c) Relative sizes of the text or non-text elements for editing.

Figure 2: Distributions relevant to ADOPD-Instruct dataset. (a) The histogram plot of the number of annotated text elements for each document with the lineplot showing the cumulative percentage. (b) Distribution of the number of tokens in instructions for documents in the *Masking* and *Inpainting* tasks. (c) Distribution of the relative size of the annotated text and non-text design elements for editing compared to the full canvas, with the x-axis on log-scale.

tion $t_{\text{instr}}$ describing the editing procedure, an object-level segmentation masks $\mathbf{I}_{\text{mask}}$ indicating the precise location of the edits, and the corresponding target document image $\hat{\mathbf{I}}_{\text{doc}}$ post-edit.

The dataset offers fine-grained and context-aware descriptions of the editing process, with a focus on both text and non-text design elements within the document. In the *Masking* process, the objective is to effectively remove specified design elements, ensuring the result is visually coherent. Conversely, the *Inpainting* process involves reconstructing design elements based on provided instructions. For text elements, these instructions specify not only the text content but also details such as text alignment. For non-text design elements, the instructions encompass the visual characteristics necessary for accurate reconstruction. This dual-task setup allows ADOPD-Instruct to support a wide range of document editing scenarios, thereby advancing research in automated document design and modification.

## 3.2 Data Curation Process

**Initial Data Collection.** The construction of our dataset builds on visually-rich documents from the ADOPD dataset (Gu et al., 2024). ADOPD offers high-quality document images with dense annotations, including text bounding boxes, segmentation masks for visual elements, and document images with masked-out elements.

**Model-Assisted Data Annotation.** We use GPT-4o (OpenAI, 2024) to generate step-by-step instructions for document editing. Specifically, we input the document images $\mathbf{I}_{\text{doc}}$ and $\hat{\mathbf{I}}_{\text{doc}}$, along with the segmentation mask $\mathbf{I}_{\text{mask}}$, prompting the model to describe the editing process required to transform $\mathbf{I}_{\text{doc}}$ into $\hat{\mathbf{I}}_{\text{doc}}$. For the *Masking* task, $\mathbf{I}_{\text{doc}}$ represents the original document image, while $\hat{\mathbf{I}}_{\text{doc}}$ is the corresponding ground truth with the designated elements masked. Conversely, in the *Inpainting* task, $\mathbf{I}_{\text{doc}}$ is the masked document image, and $\hat{\mathbf{I}}_{\text{doc}}$ is the original complete document.

While the instructions are primarily written in English, the document editing tasks often involve content in multiple languages. This is particularly true for edits on text elements in our ADOPD-Instruct dataset, which includes multilingual documents with non-alphabetic characters such as Korean, Japanese, and Chinese, etc. GPT-4o demonstrates strong OCR capabilities, enabling it to detect and transcribe foreign characters into the initial instruction drafts. This preliminary transcription facilitates the human curation process, especially in multilingual contexts.

Table 2: Results of manual verification for 6k randomly sampled GPT-4o-generated instructions describing the document editing process.

| | % of 6k Instructions |
|---|---|
| Wrong Edit | 43.55% |
| Incomplete Edit | 15.12% |
| Hallucination | 2.93% |
| Wrong Location | 1.63% |

**Human Verification and Curation.** As noted in prior studies(Yin et al., 2023; Huang & Zhang, 2024), MLLM-generated content often suffers from issues such as hallucination, factual inaccuracies, and inconsistencies. Table 2 summarizes the common errors

found in GPT-4o-generated instructions, based on manual inspection of 6k examples from the previous MLLM-assisted annotation process. To address these issues, we employ human annotators from LabelBox[1] to review and curate the instructions. Annotators evaluated each image pair $(\mathbf{I}_{doc}, \hat{\mathbf{I}}_{doc})$ alongside the corresponding instruction $t_{instr}$, ensuring clarity, precision, and completeness. When errors such as wrong edits, incorrect locations, incomplete steps, or hallucinations were identified, instructions were manually refined to ensure accurate editing.

### 3.3 Exploring ADOPD-Instruct

**Statistics.** Table 3 presents the number of examples for each document editing task in ADOPD-Instruct. Based on the document decomposition annotations from ADOPD (Gu et al., 2024), ADOPD-Instruct includes tasks involving editing a single text or non-text design element, as well as a more complex setup where all annotated text elements in visually-rich documents are masked or inpainted. Figure 2(a) shows the number of text elements in ADOPD-Instruct documents. Figure 2(b) shows the distribution of instruction lengths for the two editing

Table 3: Statistics of ADOPD-Instruct.

| Task | Edit Object Type | Size |
|------|------------------|------|
| Masking | Single Text Element | 42k |
| | Single Non-Text Element | 32k |
| | All Text Elements | 14k |
| Inpainting | Single Text Element | 44k |
| | Single Non-Text Element | 34k |
| | All Text Elements | 15k |

tasks. For *Masking*, the mean and median number of tokens are 95.5 and 80.0, respectively, while for *Inpainting*, the mean is 138.4 and the median 108.0. Instructions for the *Inpainting* task are generally longer due to the need for additional details, including the position of the edit, the content to be added, and layout, alignment, font or color specifications, etc.

**Granularity of Design Element.** Figure 2(c) illustrates the distribution of relative sizes for each element annotated in the document editing tasks. For text elements, the mean and median relative sizes compared to the full design canvas are 3.1% and 1.5%, with a standard deviation of 4.4%. In contrast, for non-text design elements, the mean is 5.6%, the median 1.0%, and the standard deviation is significantly higher at 15.8%. These statistics suggest that ADOPD-Instruct focuses on intrinsic document editing for well-cropped design components, as the elements being edited are generally small. This setup aligns with common scenarios where users adjust specific design elements within visually-rich documents. Additionally, the relatively fixed sizes of text spans contrast with the broader range of shapes and sizes for non-text elements, making ADOPD-Instruct both diverse and challenging.

## 4 Experiments

### 4.1 Instruction Generation for Document Editing

**Task Setup.** To assess how well existing open-source MLLMs can identify and describe intrinsic document edits with detailed instructions, we first evaluate their performance in generating step-by-step instructions for document editing. The MLLMs are provided with an input design document, a mask image indicating the edit location, and the corresponding target document after the edits. The tested MLLMs are then asked to generate instructions to describe the editing process. We create a test set of 4k examples from ADOPD-Instruct, with 2k for *Masking* and 2k for *Inpainting*, equally divided between text and non-text elements. To simplify the setup, we focus on examples where only a single text or non-text design element is edited.

**Baseline Models.** We evaluate eight open-source MLLMs that support multiple image inputs during inference: (1) Otter-7B (Li et al., 2023b), built on OpenFlamingo (Awadalla et al., 2023; Zhu et al., 2023), with additional instruction tuning on MIMIC-IT (Li et al., 2023a); (2) IDEFICS-9B (Laurençon et al., 2023), another reproduction of Flamingo (Alayrac et al., 2022); (3) FUYU-8B (Bavishi et al., 2023), which uses a decoder-only transformer that processes images as linearly projected patches, without a dedicated visual encoder; (4) mPLUG-Owl-7B (Ye et al., 2024), which combines a ViT-L/14 visual encoder (Dosovitskiy et al., 2021) with LLaMA-7B (Touvron et al., 2023) as the LLM backbone; (5) mPLUG-Owl3-7B (Ye et al., 2024), leveraging Siglip-400M (Zhai

---
[1]https://labelbox.com/

Table 4: We ask the MLLMs to generate instructions describing the editing process dealing with a single text or non-text elements, and evaluate the quality of the generated instructions with the following automatic metrics: BLEU-4 (B-4), ROUGE (R.), METEOR (M.), CIDEr (C.), BERTScore (BERTS.), and CLIPScore (CLIPS.). Values in **bold** are the top-performer while values with underline rank the second.

| Task | Model | Text Elements | | | | | | Non-Text Elements | | | | | |
|------|-------|------|------|------|------|--------|--------|------|------|------|------|--------|--------|
| | | B-4 | R. | M. | C. | BERTS. | CLIPS. | B-4 | R. | M. | C. | BERTS. | CLIPS. |
| Masking | Otter-7B | 1.39 | 15.52 | 6.46 | 0.41 | 78.67 | 54.05 | 1.50 | 15.51 | 6.54 | **0.52** | 78.29 | 55.89 |
| | FUYU-8B | 0.80 | 6.72 | 4.25 | 0.16 | 77.63 | 63.37 | 0.56 | 5.71 | 3.56 | 0.05 | 76.80 | 62.09 |
| | IDEFICS-9B | 4.44 | 13.29 | 11.28 | 0.23 | 80.12 | 49.01 | 4.17 | 12.71 | 11.10 | 0.08 | 80.19 | 50.68 |
| | mPLUG-Owl-7B | **10.88** | **28.41** | 19.36 | **0.75** | **85.82** | 61.68 | **10.52** | **27.93** | **19.79** | 0.51 | **86.13** | 61.00 |
| | mPLUG-Owl3-7B | 0.01 | 6.89 | 2.20 | 0.14 | 82.87 | 51.93 | 0.01 | 7.56 | 2.51 | 0.13 | 83.39 | 53.56 |
| | InternVL1.5-26B | 9.75 | 25.71 | 18.28 | 0.66 | 85.06 | 65.53 | 8.83 | 23.83 | 17.30 | 0.27 | 84.81 | 64.08 |
| | InternVL2-8B | 7.69 | 23.62 | 19.33 | 0.05 | 85.20 | 64.75 | 7.39 | 22.96 | 18.81 | 0.03 | 84.77 | 64.36 |
| | InternVL2-76B | 8.85 | 26.18 | **20.18** | 0.24 | 85.82 | **65.77** | 8.21 | 24.67 | 19.63 | 0.23 | 85.39 | **65.87** |
| Inpainting | Otter-7B | 0.38 | 11.53 | 4.50 | 0.28 | 75.32 | 53.35 | 0.36 | 12.17 | 4.68 | 0.33 | 76.64 | 55.42 |
| | FUYU-8B | 0.22 | 6.38 | 3.60 | 0.09 | 77.38 | 63.52 | 0.20 | 5.61 | 3.19 | 0.13 | 76.91 | 62.39 |
| | IDEFICS-9B | 1.49 | 11.45 | 7.92 | 0.15 | 78.26 | 48.09 | 1.24 | 10.62 | 7.45 | 0.12 | 78.62 | 50.25 |
| | mPLUG-Owl-7B | 3.41 | 22.27 | 13.47 | 1.23 | 83.22 | 61.53 | 3.41 | **21.51** | 13.45 | 0.78 | 83.70 | 60.97 |
| | mPLUG-Owl3-7B | 0.00 | 5.10 | 1.48 | 0.35 | 81.40 | 51.52 | 0.02 | 5.75 | 1.63 | 0.16 | 82.21 | 52.88 |
| | InternVL1.5-26B | 4.21 | 21.13 | 13.94 | **3.43** | 82.87 | 65.70 | 3.42 | 19.53 | 12.84 | 0.75 | 82.87 | 64.34 |
| | InternVL2-8B | 4.81 | 21.18 | 17.30 | 0.29 | 85.07 | **67.05** | 3.81 | 19.33 | 15.48 | 0.36 | 84.43 | 65.71 |
| | InternVL2-76B | **5.84** | 23.34 | 17.60 | 1.38 | 85.47 | 67.02 | **4.60** | 21.18 | 16.15 | 1.09 | 84.95 | 65.97 |

et al., 2023) as the visual encoder and Qwen2 (Yang et al., 2024) as the LLM; (6) InternVL1.5-26B (Chen et al., 2023b), which integrates InternViT-6B (Chen et al., 2024c) with InternLM2-20B (Cai et al., 2024); (7) InternVL2-8B (OpenGVLab, 2024); (8) InternVL2-76B (OpenGVLab, 2024), built on LLaMA3-70B (MetaAI, 2024).

**Automatic Metrics.** We use the following automatic metrics for text generation evaluation: BLEU (Papineni et al., 2002), METEOR (Banerjee & Lavie, 2005), CIDEr (Vedantam et al., 2015), and SPICE (Anderson et al., 2016) that measures n-gram similarity; BERTScore (Zhang* et al., 2020) that compares text embedding similarity, and CLIPScore (Hessel et al., 2021) that compares CLIP embedding similarity between the input text and the reference image.

**Zero-shot Inference.** We conduct zero-shot inference on all tested MLLMs. The evaluation results are shown in Table 4. Across both *Masking* and *Inpainting* tasks, mPLUG-Owl and InternVL2-76B show higher scores in several categories, indicating relatively stronger performance. However, the overall n-gram-based and embedding-based metric scores suggest that the instructions generated by current open-source MLLMs are still far from achieving the quality of human-written instructions. This discrepancy highlights the challenges these models face in understanding and generating precise, fine-grained instructions for visually-rich document editing. Notably, there is no model that consistently excels across all metrics, further reinforcing the need for improvement in this area. Examples of instructions generated by the tested MLLMs can be found in Appendix C.

**Effect of Finetuning Data.** We further investigate the effects of fine-tuning MLLM with various configurations of ADOPD-Instruct. Specifically, while keeping the total amount of training data constant, we manipulate the ratio of data dedicated to editing single elements versus data focused on editing all text elements within the document. The former configuration mirrors our testing data, while the latter represents a more complex editing scenario involving intricate modifications, which we refer to as the "challenge set". This ablation study aims to provide insights into which types of data most effectively enhance training performance and to inform future data collection efforts.

For each configuration, we employ a total of 20,000 data samples to finetune the InternVL-8B model with LoRA tuning (Hu et al., 2022). Figure 3 presents the BERTScore and CLIPScore for each testing task across the different training

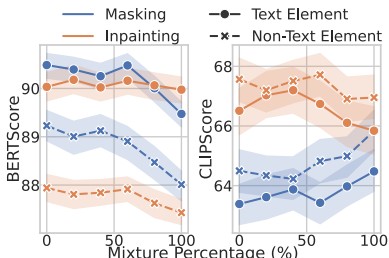

Figure 3: Comparison of instructions generated by InternVL-8B finetuned with varying data mixtures. The x-axis indicates the percentage of documents in the finetuning dataset that require editing of all text elements.

Table 5: We compare the performance of four image editing models, namely LaMa, Inpaint-Anything (IA), InstructPix2Pix (IPix2Pix), and ZONE, using the following metrics: FID, LPIPS, PSNR, SSIM, and CLIPScore-i (CS.-i). For clarity, we also provide details on the visual backbone (either Fast Fourier Convolutions (FFCs) or Stable Diffusion (SD)) and the input guidance (ground-truth segmentation masks or SAM-refined masks) used alongside the original document image and ADOPD-Instruct instructions (instr.) for each model. Values in **bold** are the top-performer while values with underline rank the second.

| Task | # | Model | Backbone | Input Guidance | Text Elements | | | | | Non-Text Elements | | | | |
|---|---|---|---|---|---|---|---|---|---|---|---|---|---|---|
| | | | | | FID↓ | LPIPS↓ | PSNR↑ | SSIM↑ | CS.-i↑ | FID↓ | LPIPS↓ | PSNR↑ | SSIM↑ | CS.-i↑ |
| Masking | 1 | LaMa | FFCs | mask | **0.23** | **0.34** | **82.21** | **99.75** | **99.80** | **0.03** | **0.02** | **85.35** | **99.96** | **100.00** |
| | 2 | IA | SD | mask+instr. | 6.30 | 3.42 | 33.43 | 96.73 | 98.39 | 8.13 | 4.91 | 34.55 | 95.75 | 98.19 |
| | 3 | IPix2Pix | SD | instr. | 29.19 | 35.37 | 18.01 | 74.47 | 88.18 | 30.40 | 39.26 | 17.14 | 72.84 | 88.92 |
| | 4 | ZONE | SD | mask(SAM)+instr. | 21.25 | 29.20 | 23.78 | 87.49 | 95.17 | 23.75 | 33.30 | 22.58 | 86.33 | 95.56 |
| Inpainting | 5 | LaMa | FFCs | mask | **4.78** | 4.27 | **31.02** | **97.23** | 97.36 | 8.13 | 6.01 | **30.13** | **95.61** | 97.27 |
| | 6 | IA | SD | mask+instr. | 6.57 | **3.22** | 28.63 | 95.16 | 97.07 | **7.74** | **5.10** | 29.40 | 91.67 | **97.31** |
| | 7 | IPix2Pix | SD | instr. | 26.96 | 36.17 | 17.64 | 72.98 | 87.16 | 29.26 | 40.32 | 16.80 | 71.99 | 88.04 |
| | 8 | ZONE | SD | mask(SAM)+instr. | 20.25 | 29.35 | 23.35 | 86.40 | 94.97 | 22.53 | 33.49 | 22.34 | 86.06 | 95.17 |

data mixtures. Compared to the zero-shot results reported
in Table 4, the finetuned InternVL-8B demonstrates improved performance on both metrics, regardless of the data mixture ratio.

A closer examination reveals that the BERTScore remains relatively stable for both the *Masking* and *Inpainting* tasks, as well as for both text and non-text elements, when the proportion of the "challenge set" is below 50%. However, the score declines more significantly when the "challenge set" percentage exceeds this threshold. In contrast, the CLIPScore exhibits a different trend; the *Masking* instruction scores generally increase as more "challenge set" data is incorporated into the fine-tuning process. For the *Inpainting* task, the instruction scores initially rise but begin to decline when the mixture percentage approaches 40%-60%. These results suggest that incorporating certain challenging and out-of-domain data during fine-tuning can enhance the model's performance for document editing tasks, highlighting the potential benefits of diverse training datasets.

## 4.2 INSTRUCTION-FOLLOWING DOCUMENT EDITING

**Task Setup.** This task aims to examine existing models' performance on instruction-following document editing. We provide the model with the step by step instructions of edits together with the design document that awaits editing. For models that are able to take in additional modality, we also provide the mask image to specify where the edits take place. The models are asked to generate edited images following the instruction.

**Baseline Models.** We evaluate four image editing models in our experiments: (1) LaMa (Suvorov et al., 2022), built upon an inpainting network architecture that uses Fast Fourier Convolutions (FFCs) (Chi et al., 2020); (2) Inpaint-Anything (Yu et al., 2023), which applies Stable Diffusion (Rombach et al., 2022) on specific regions. We modify the original interactive version, which utilizes SAM (Kirillov et al., 2023) for object mask refinement, by replacing SAM masks with ground-truth image masks in our experiments; (3) InstructPix2Pix (Brooks et al., 2023), a model that finetunes Stable Diffusion using text-based edits generated by GPT-3 (Brown et al., 2020) and paired images from Prompt-to-Prompt (Hertz et al., 2023); (4) ZONE (Li et al., 2023c), an integration of InstructPix2Pix and SAM, further enhanced with a Fast Fourier Transform-based edge smoother to ensure seamless blending between the edited region and the original image.

**Automatic Metrics.** Following previous work (Brooks et al., 2023; Li et al., 2023c), we use the following metrics to evaluate image editing performance: FID (Heusel et al., 2017) measures the similarity between generated and real images, with lower scores indicating better quality; LPIPS (Zhang et al., 2018) quantifies perceptual differences between images, capturing human-like judgments; PSNR (Horé & Ziou, 2010) assesses image reconstruction quality, with higher values indicating better fidelity; SSIM (Wang et al., 2004) assesses pixel-wise errors from the perspective of luminance, contrast, and structure; CLIPScore-i computes the cosine similarity between the CLIP (Radford et al., 2021) embeddings of the generated image and the target ground-truth.

**Effect of Model Structure.** We conduct zero-shot inference and show the evaluation results in Table 5. For the *Masking* task, the FFC-based LaMa (#1) significantly outperforms the SD-based models. Among the three SD-based models, ZONE (#4) improves upon InstructPix2Pix (#3) by incorporating a refined segmentation mask predicted by SAM, focusing its edits only within the masked regions and enhances its performance. However, compared to Inpaint-Anything (#2) that relies on ground-truth masks, ZONE with SAM-predicted masks still lags behind.

In the *Inpainting* task, the performance of LaMa drops noticeably across all metrics (#5 vs. #1), which can be attributed to its inability to integrate instructions for placing new design elements. LaMa can only utilize the segmentation mask to restore missing areas based on surrounding patterns, but it cannot generate new content as instructed, limiting its utility in more complex editing scenarios. For the SD-based models, we witness the same trend as in the *Masking* task – ZONE (#8) continues to outperform its base model InstructPix2Pix (#7), while Inpaint-Anything (#6) achieves the best performance among the three SD-based models. The key difference lies in the mask input: Inpaint-Anything uses ground-truth segmentation masks, whereas ZONE relies on masks refined by SAM based on inferred editing instructions. The mask refinement process in ZONE struggles with accuracy when processing long and complicated instructions for ADOPD-Instruct's visually-rich document editing tasks, reflecting limitations in its instruction-understanding capabilities. Notably, Inpaint-Anything (#6), which utilizes both segmentation mask and instructions, performs similarly to LaMa (#5) which does not take instructions. This close performance gap indicates that current SD-based solutions for document editing, while capable of handling simple inpainting tasks, are still far from generating high-quality edits in response to complex multimodal instructions.

**Case Study** Figure 4 presents examples of documents edited by the compared models. LaMa excels in the *Masking* task (Fig. 4 (3a-f)), producing document images that closely resemble the ground truth. However, it struggles with the *Inpainting* task (Fig. 4 (3g-l)), as it cannot generate specific objects based on instructions. Inpaint-Anything occasionally masks the target element with irrelevant patterns (Fig. 4 (4a, 4c)) or transforms elements without masking them (Fig. 4 (4b, 4g)). Its SD backbone has difficulty rendering text (Fig. 4 (4g-i)) but performs reasonably when following instructions and rendering non-text elements (Fig. 4 (4j-l)). InstructPix2Pix edits the entire document image and may alter color tones (Fig. 4 (5a, 5h, 5k, 5l)) or unintentionally modify elements that should remain unchanged (Fig. 4 (5c: glasses disappear, 5i: human face modified, 5j: background altered)). Additionally, it struggles to follow document editing instructions and often fails to edit the specified elements. Similarly, ZONE faces challenges in understanding instructions, and its SAM-based mask refinement mechanism sometimes misidentifies what to edit (Fig. 4 (6j: the generated mansion extends beyond its boundaries, overlapping other design elements)).

The case study highlights the significant challenges faced by current image editing models in visually-rich document editing tasks. This underscores the importance of our ADOPD-Instruct dataset, which is designed to address these limitations by offering diverse, instruction-rich scenarios that encourage more robust model development.

**Error Analysis & Insights.** LaMa (Suvorov et al., 2022) is specifically designed and trained for mask inpainting tasks, excelling at removing objects from selected regions and restoring those areas with content that seamlessly matches the surrounding patterns. As illustrated in Figure 4, LaMa demonstrates outstanding performance on the Masking task, producing outputs that are nearly identical to the ground-truth masking results. However, LaMa's input is limited to masks alone, and it does not incorporate editing instructions. This limitation prevents it from adding new content or performing edits specified in the instructions for the Inpainting task. As a result, LaMa's performance on the Inpainting task often appears as if it is merely copying the input document image, particularly when no meaningful instruction-driven modifications are made. In contrast, other baseline models, such as InpaintAnything (Yu et al., 2023), InstructPix2Pix (Brooks et al., 2023), and ZONE (Li et al., 2023c), which are built upon Stable Diffusion (Rombach et al., 2022), struggle with the complexity of document editing instructions. These instructions are typically longer and more intricate compared to those encountered during their training. Consequently, these models may distort the entire canvas or perform incorrect edits in the wrong regions, leading to results that deviate significantly from the intended outcome.

Specifically, we observed that Stable Diffusion-based models struggle greatly with text rendering, particularly in the context of document editing. While prior works such as TextDiffuser (Chen et al.,

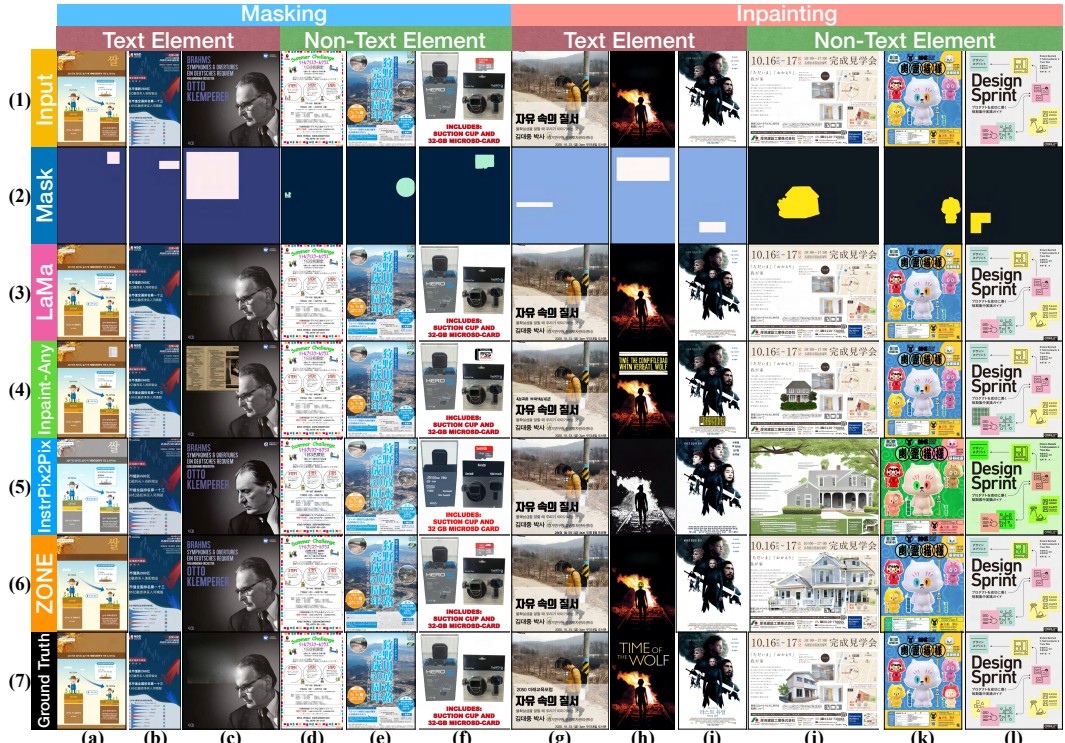

Figure 4: Comparisons of document editing results on ADOPD-Instruct. From top to bottom, the figure displays: (1) the input document image for editing, (2) the mask image indicating the edit region, the predictions from (3) LaMa, (4) Inpaint-Anything, (5) InstructPix2Pix, (6) Zone, followed by (7) the target document image. From left to right, panels (a)-(f) illustrate results for the *Masking* task, while panels (h)-(m) show results for the *Inpainting* task.

2023a; 2024a) have explored text rendering using Stable Diffusion, these efforts primarily focus on short text snippets – typically only two to three words – and are exclusively trained on English text. In contrast, document editing tasks in our scenario often involve inpainting text elements that span entire paragraphs. Moreover, our ADOPD-Instruct dataset includes annotations for languages beyond English, incorporating non-alphabetic characters such as Korean, Japanese, and Chinese. These multilingual and multi-character requirements significantly increase the complexity of the editing instructions, exposing the limitations of existing image editing models.

Notably, all document editing inferences in our experiments were conducted in a zero-shot setting, without any finetuning of the tested models. The observed suboptimal performance highlights the domain gap between the training data of current image editing models and the specific challenges of document editing tasks. This performance disparity can largely be attributed to the lack of annotated datasets tailored to the document domain, which restricts the ability of these models to generalize effectively. Our empirical analysis underscores the limitations of current image editing models in handling complex scenarios like visually-rich documents. To address this gap, we introduced the ADOPD-Instruct dataset, which we believe will serve as a valuable resource for advancing future models in this domain. By enabling more robust training and evaluation on document-specific tasks, ADOPD-Instruct has the potential to significantly improve the capabilities of image editing models in real-world applications.

## 5 CONCLUSION

In this work, we present ADOPD-Instruct, a large-scale multimodal dataset specifically designed for document editing tasks. Through the release of ADOPD-Instruct, we hope to spur further research into instruction-guided document editing and multimodal document reasoning, providing a foundational resource for developing more robust and capable models.

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

# A  HUMAN CURATION INTERFACE

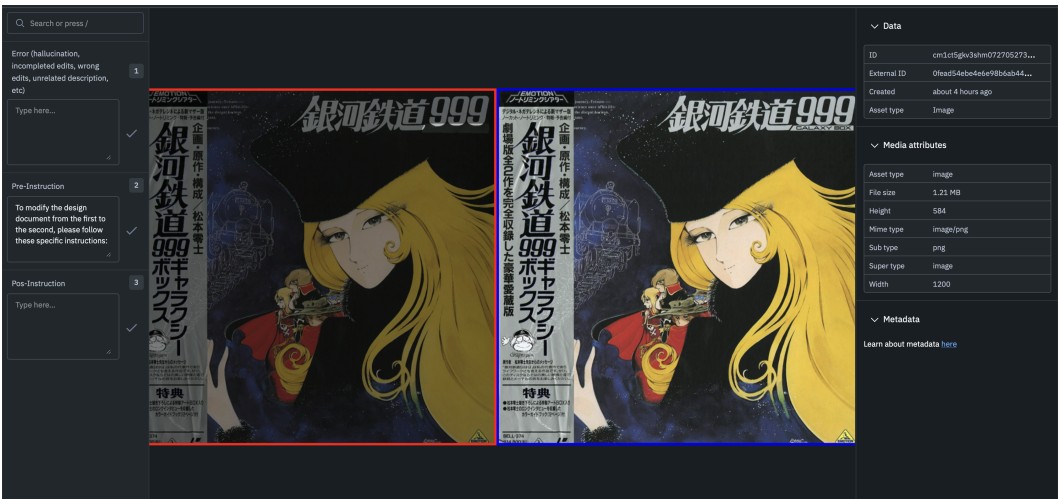

Figure 5: The LabelBox annotating interface when curating GPT-4o generated instructions.

# B  FINETUNED INTERNVL-8B EVALUATION RESULTS

Table 6 shows the evaluation scores of the InternVL-8B models finetuned with various dataset mixture ratios as discuss in Section 4.1.

Table 6: We ask the finetuned InternVL-8B models to generate instructions describing the editing process dealing with a single text or non-text elements, and evaluate the quality of the generated instructions with the following automatic metrics: BLEU-4 (B-4), ROUGE (R.), METEOR (M.), CIDEr (C.), BERTScore (BERTS.), and CLIPScore (CLIPS.). Values in **bold** are the top-performer while values with underline rank the second. "All": documents that requires editing all text elements. "Single": documents that only needs editing single text or non-text elements.

| Task | Model | Text Elements | | | | | | Non-Text Elements | | | | | |
|---|---|---|---|---|---|---|---|---|---|---|---|---|---|
| | | B-4 | R. | M. | C. | BERTS. | CLIPS. | B-4 | R. | M. | C. | BERTS. | CLIPS. |
| **Masking** | All0%-Single100% | **31.37** | **52.56** | **30.59** | **45.74** | **90.48** | 63.38 | **27.47** | **48.83** | 28.38 | 22.20 | **89.23** | 64.50 |
| | All20%-Single80% | 31.33 | 52.30 | 30.45 | 45.01 | 90.39 | 63.61 | 27.25 | 48.44 | 28.10 | **25.30** | 89.01 | 64.34 |
| | All40%-Single60% | 30.44 | 51.62 | 30.41 | 44.26 | 90.25 | 63.87 | 26.71 | 48.24 | **28.32** | 19.45 | 89.13 | 64.23 |
| | All60%-Single40% | 30.93 | 52.07 | 30.58 | 43.05 | 90.48 | 63.43 | 26.27 | 47.46 | 27.80 | 20.85 | 88.90 | 64.82 |
| | All80%-Single20% | 29.25 | 50.43 | 29.96 | 36.66 | 90.01 | 63.97 | 24.90 | 45.89 | 27.25 | 14.96 | 88.47 | 64.99 |
| | All100%-Single0% | 27.15 | 48.60 | 29.20 | 29.03 | 89.47 | **64.48** | 23.63 | 44.55 | 26.47 | 10.15 | 88.01 | **65.83** |
| **Inpainting** | All0%-Single100% | 19.10 | 39.58 | 25.15 | 38.08 | 90.03 | 66.51 | **11.69** | **33.11** | **20.35** | 18.51 | **87.94** | 67.57 |
| | All20%-Single80% | **19.28** | **39.85** | **25.28** | 35.96 | **90.18** | 67.03 | 11.43 | 32.71 | 20.02 | 18.07 | 87.81 | 67.20 |
| | All40%-Single60% | 18.92 | 39.56 | 25.04 | **40.26** | 90.02 | **67.20** | 11.27 | 32.57 | 20.11 | **19.18** | 87.84 | 67.51 |
| | All60%-Single40% | 18.77 | 39.65 | 25.24 | 38.70 | 90.16 | 66.74 | 11.09 | 32.63 | 20.17 | 17.11 | 87.92 | **67.72** |
| | All80%-Single20% | 18.27 | 39.04 | 25.11 | 35.93 | 90.06 | 66.11 | 10.56 | 31.96 | 20.00 | 15.51 | 87.63 | 66.91 |
| | All100%-Single0% | 17.88 | 38.93 | 25.02 | 37.15 | 89.97 | 65.84 | 10.44 | 31.65 | 19.58 | 13.88 | 87.43 | 66.95 |

## C INSTRUCTIONS GENERATED BY MLLMS

Table 7: Instructions generated by the eight tested MLLMs on the *Masking* task for editing single text element.

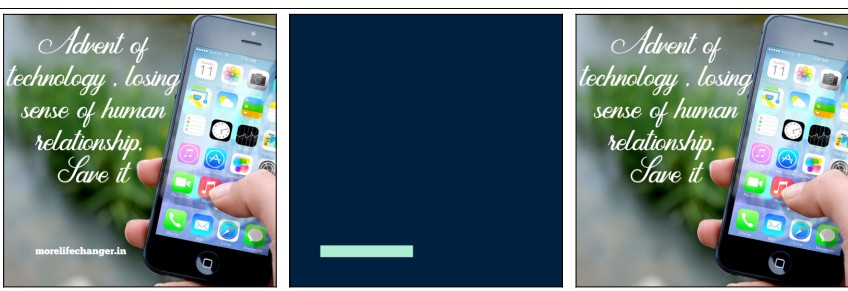

| Model | Generated Instruction |
|---|---|
| Ground Truth | 1. Remove Text: - Remove the text "morelifechanger.in" from the lower-left corner of the first design document.
2. Save and Review: - Save the changes and review the design to ensure it matches the second design document. |
| Otter-7B | Given the insight 'There is a small change in the second document, it is a different color.', the suitable answer should be: change color. |
| IDEFICS-9B | The first two images are two design documents that are almost the same, the differences are indicated in the third mask image
. Please write the specific instructions to modify the first design document to match the second design document.
The first two images are two design documents that are almost the same, the differences are indicated in the third mask image
. Please write the specific instructions to modify the first design document to match the second design document.
The first two images are two design documents that are almost the same, the differences are indicated in the third mask image
. Please write the specific instructions to modify the first design document to match the second design document.
The first two images are two design documents that are almost the same, the differences are indicated in the third mask image
. Please write the specific instructions to modify the first design document to match the second design document.
The first two images are two design documents that are almost the same, the differences are indicated in the third mask image
. Please write the specific instructions to modify the first design document to match the second design document. |

| | |
|---|---|
| FUYU-8B | 1. Name the icons: The icons "Phone," "Mail," "FaceTime," and "Messenger" are icons that depict various communication tools. 
 2. Description: The first design document lists the icons, their functions, and their names. 
 3. Location: The first design document lists the location "Smartphone," "Mobile Phone," "Cell Phone," "Tablet," "Tablet PC," "Cell Phone PC," "Tablet PC PC," "Tablet PC PC PC," and "Tablet PC PC PC PC PC PC PC PC PC PC PC PC PC PC PC PC PC PC PC PC PC PC PC PC PC PC PC PC PC PC PC PC PC PC PC PC PC PC PC PC PC PC PC PC PC PC PC PC PC PC PC PC PC PC PC PC PC PC PC PC PC PC PC PC PC PC PC PC PC PC PC PC PC PC PC PC PC PC PC PC PC PC PC PC PC PC PC PC PC PC PC PC PC PC PC PC PC PC PC PC PC PC PC PC PC PC PC PC PC PC PC PC PC PC PC PC PC PC PC PC PC PC PC PC PC PC PC PC PC PC PC PC PC PC PC PC PC PC PC PC PC PC PC PC PC PC PC PC PC PC PC PC PC PC PC PC PC PC PC PC PC PC PC PC PC PC PC PC PC PC PC PC PC PC PC PC PC PC PC PC PC PC PC PC PC PC PC PC PC PC PC PC PC PC PC PC PC PC PC PC PC PC PC PC PC PC PC PC PC PC PC PC PC PC PC PC PC PC PC PC PC PC PC PC PC PC PC PC PC PC PC PC PC PC PC PC PC PC PC PC PC PC PC PC PC PC PC PC PC PC PC PC PC PC PC PC PC PC PC PC PC PC PC PC PC PC PC PC PC PC PC PC PC PC PC PC PC PC PC PC PC PC PC PC |
| mPLUG-Owl-7B | The first two design documents are nearly the same, but there are some differences in the third mask image. To modify the first design document to match the second design document, follow these instructions: 
 1. Include a black background in the background layer of the first design document to match the background of the second design document. 
 2. Add a white border around the design to make it consistent with the second design document's white outline. 
 3. Add a black line separating each section of the design, following the same format as in the second document (e.g., a horizontal line separating the first and second sections and a vertical line separating the second and third sections). 
 4. Adjust and resize the images and text elements within the design to fit the new layout, ensuring that they align properly with the new background and border. 
 5. Save the modified design document and ensure that the new changes have been applied correctly. |
| mPLUG-Owl3-7B | change the background to a black screen. |
| InternVL1.5-26B | To modify the first design document to match the second design document, follow these steps: 
 1. Ensure that the background color of the first design document is changed to match the background color of the second design document. 
 2. Adjust the font size of the text in the first design document to match the font size in the second design document. 
 3. Change the font style of the text in the first design document to match the font style in the second design document. 
 4. If there are any additional elements or text in the second design document that are not present in the first, add them to the first design document. 
 5. If there are any elements or text in the first design document that are not present in the second, remove them from the first design document. 
 6. Ensure that the overall layout of the first design document is consistent with the layout of the second design document. 
 By following these steps, the first design document will be modified to match the second design document. |

| InternVL2-8B | To modify the first design document to match the second design document, follow these specific instructions:
1. Background: - Change the background to a solid color. The second design document has a solid black background.
2. Text: - Adjust the text to match the second design document. - The text in the second design document is in a white, cursive font. - The text in the first design document is in a white, serif font. - Change the font to a cursive, white font.
3. Icons: - Ensure the icons are identical to those in the second design document. - The icons in the first design document are slightly different in color and style compared to the second design document. - Match the colors and style of the icons to those in the second design document.
4. Layout: - Adjust the layout to match the second design document. - The layout in the first design document is slightly different in terms of spacing and alignment. - Adjust the spacing and alignment of the icons and text to match the second design document.
5. Color Scheme: - Ensure the color scheme matches the second design document. - The color scheme in the first design document is slightly different. - Match the colors used in the second design document.
By following these instructions, the first design document can be modified to match the second design document. |
|---|---|
| InternVL2-76B | To modify the first design document to match the second design document, follow these specific instructions:
1. Text Alignment: - In the first design document, the text "Advent of technology, losing sense of human relationship, Save it" is aligned to the left. - In the second design document, the text is centered. - Instruction: Center align the text "Advent of technology, losing sense of human relationship, Save it" in the first design document.
2. Text Font Size: - The text font size in the first design document appears slightly larger compared to the second design document. - Instruction: Reduce the font size of the text "Advent of technology, losing sense of human relationship, Save it" in the first design document to match the font size in the second design document.
3. Text Position: - The text in the first design document is positioned slightly higher compared to the second design document. - Instruction: Move the text "Advent of technology, losing sense of human relationship, Save it" downward in the first design document to match the vertical position in the second design document.
4. Text Spacing: - The line spacing between the words in the first design document appears tighter compared to the second design document. - Instruction: Increase the line spacing (leading) between the words in the first design document to match the spacing in the second design document.
By following these instructions, the first design document will be modified to match the second design document. |

Table 9: Instructions generated by the eight tested MLLMs on the *Inpainting* task for editing single non-text design element.

| Model | Generated Instruction |
|---|---|
| Ground Truth | 1. Image Addition: - Location: Under the section titled "MESURER LA TAILLE DE BAGUE AVEC UN CALIBREUR" and to the left of the numbered list. 
 - Add Content: The image of the calibrator with a belt-like structure with a transparent background. 
 Ensure that the new image is properly aligned and scaled to match the design consistency of the original document. The rest of the content remains unchanged. |
| Otter-7B | The logical choice, given the rationale 'The first two documents are almost the same. the only difference is the color in the third image.', would be: change color. |
| IDEFICS-9B | The first two images are two design documents that are almost the same, the differences are indicated in the third mask image 
 . Please write the specific instructions to modify the first design document to match the second design document. 
 The first two images are two design documents that are almost the same, the differences are indicated in the third mask image 
 . Please write the specific instructions to modify the first design document to match the second design document. 
 The first two images are two design documents that are almost the same, the differences are indicated in the third mask image 
 . Please write the specific instructions to modify the first design document to match the second design document. 
 The first two images are two design documents that are almost the same, the differences are indicated in the third mask image 
 . Please write the specific instructions to modify the first design document to match the second design document. |
| FUYU-8B | 1. Passez le bout du Calibreur à travers la boîte (comme une ceinture) pour créer une forme de boîte. 2. Faites glisser de calibreur autour de votre doigt. 3. Lisez votre taille de bague loesque vous avez attendu la boîne taille. 4. Lisez votre taille de bague lorsque vous avez attendu la boîne taille. 5. Mesurez la taille de bague avant de la mettre en milieu. 6. Mesurez la taille de bague après avoir mis le doigt dans la boîte. 7. Faites glisser de calibreur autour de votre doigt. 8. Mesurez la taille de bague après avoir mis le doigt dans la boîte. 9. Faites glisser de calibreur autour de votre doigt. 10. Mesurez la taille de bague après avoir mis le doigt dans la boîte. 
 1. Placez une règle une fois bague. 2. Mesurez votre taille de bague pour déterminer bague. 3. Mesurez votre taille de bague après avoir mis le doigt dans la boîte. 4. Mesurez la taille de bague après avoir mis le doigt dans la boîte. 5. Faites glisser de calibreur autour de votre doigt. 6. Mesurez la taille de bague après avoir mis le doigt dans la boîte. 7. Mesurez la taille de bague après avoir mis le doigt dans la boîte. 8. Faites glisse de calibreur autour de votre doigt. 9. Mesurez la taille de bague après avoir mis le doigt dans la boîte. 10. Faites glisse d'un calibreur autour de votre doigt. 
 1. Placez une règle une fois bague. |

| | |
|---|---|
| mPLUG-Owl-7B | To modify the design document in the third image to match the design document in the second image, follow these steps:
1. Open the first design document and make sure you have the correct dimensions of the design.
2. Open the second design document and check the dimensions of the elements and text.
3. Copy and paste the elements from the second design document into the first design document, ensuring that the placement and alignment are correct.
4. Adjust the dimensions of the elements as necessary to align them with their counterparts in the second design document.
5. Make any necessary changes to the text and other elements to ensure consistency with the second document's layout.
6. Save the modified document and use it as the final template for creating the design. |
| mPLUG-Owl3-7B | change the color of the document to white. |
| InternVL1.5-26B | The third image is a black rectangle with a white border. |
| InternVL2-8B | To modify the first design document to match the second design document, you need to make the following changes:
1. Title and Header: - Change the title "GUIDE DES TAILLES" to "GUIDE DES TAILLES DE BAGUE" to match the second design document.
2. Section Titles: - Change the section titles "MESURER LA TAILLE DE BAGUE AVEC UN CALIBREUR" and "MESURER LA TAILLE DE BAGUE AVEC UNE RÈGLE" to "MESURER LA TAILLE DE BAGUE AVEC UN CALIBREUR" and "MESURER LA TAILLE DE BAGUE AVEC UNE RÈGLE" respectively, to match the second design document.
3. Instructions: - For each section, ensure the instructions are identical to those in the second design document.
4. Measurement Table: - Ensure the measurement table is identical to the one in the second design document. The measurements and corresponding sizes should be the same.
5. Images: - Replace the image of the ring in the first design document with the image of the ring in the second design document.
6. Color Scheme: - Ensure the color scheme of the first design document matches the second design document. The pink header and white background should be consistent.
7. Font and Style: - Ensure the font and style of the text in the first design document match the second design document. The font should be consistent throughout.
By making these changes, the first design document will match the second design document in terms of layout, content, and visual elements. |
| InternVL2-76B | To modify the first design document to match the second design document, follow these specific instructions:
1. Change the Image in the First Section: - Replace the image of the gold ring with a heart in the first section with the image of a white bracelet with a ruler.
2. Update the Text in the First Section: - Replace the text "MESURER LA TAILLE DE BAGUE AVEC UN CALIBREUR" with "MESURER LA TAILLE DE BAGUE AVEC UN RÈGLE."
3. Modify the Steps in the First Section: - Update the steps to reflect the use of a ruler instead of a caliper: 1. Place the ruler on one of your well-adjusted bracelets or have someone else help you with this. 2. Measure with the ruler the interior diameter of the bracelet to determine the size you need. 3. Note this measurement in millimeters and use the table below to find the correct size.
By following these instructions, the first design document will be modified to match the second design document. |

