# OpenReview forum: "ADOPD-Instruct: A Large-Scale Multimodal Dataset for Document Editing"
_ICLR.cc/2025/Conference — Submitted to ICLR 2025_

### Official Review · Reviewer_oGDA · 2024-10-31

**Soundness:** 2
**Presentation:** 2
**Contribution:** 2
**Rating:** 5
**Confidence:** 5

**Summary:**

The paper builds the large-scale multimodal dataset with entity-level annotations and step-by-step instructions for visually-rich document editing. They evaluate the performance of MLLMs in visually-rich document understanding, as well as the efficacy of leading image editing models in document editing tasks.

**Strengths:**

1. They introduce the publicly available multimodal dataset with detailed annotations and step-by-step instructions for entity-level editing in visually-rich documents. Builting upon ADOPD documents, they first use GPT-4o to generate initial editing instructions, which are then refined by human annotators to ensure accuracy and validity.
2. Experimental results show that the instructions generated by current open-source MLLMs did not fully achieve the level of detail and precision found in human-written instructions when describing intricate edits between visually-rich document.
3. They found the limitations of current MLLMs and image editing models in performing visuallyrich document editing tasks, emphasizing the necessity for more sophisticated methodologies to enhance model performance in this domain.

**Weaknesses:**

1. The evaluation metrics are not comprehensive enough, for example, they could be based on GPT-4.
2. There is a lack of more insightful analysis, such as whether newer models perform worse than older ones, and the reasons behind the differences among various models.
3. Lacking some error analysis and insights to optimize the model's performance.

**Questions:**

see weaknesses.

---

> ### Author Response · Authors · 2024-11-27
> **Author Response to Reviewer oGDA (Part 1)**
>
> > **Evaluation based on GPT-4**
>
> We sincerely thank the reviewer for the thoughtful suggestion. In our initial submission, we chose to evaluate the instruction generation results using widely adopted automatic metrics for text generation and to assess instruction-following document editing results with commonly used image similarity metrics. These are standard practices in text and image editing evaluation.
>
> We did not initially use GPT-4 for evaluating the alignment between ground-truth instructions and model-generated ones because, although the ground-truth instructions were curated by humans, their foundation was generated by GPT-4. This raised concerns about potential biases, as GPT-4 might exhibit a preference for its own generated instructions, potentially leading to an unfair evaluation.
>
> That said, we recognize the potential value of leveraging GPT-4 in the evaluation process. For instance, GPT-4 could be utilized in a reference-free pairwise comparison of outputs generated by different baseline models. This could provide additional insights and complement our current evaluation approach. We will incorporate these considerations into our future work to strengthen the evaluation process. Thank you for the opportunity to reflect and improve upon this aspect of our study.
>
>
> > **More Analysis on Experimental Results**
>
> We thank the reviewer for the thoughtful suggestion! In response, we have added the following analysis of the experimental results to the revised draft (please see page 8~9):
>
> LaMa is specifically designed and trained for mask inpainting tasks, excelling at removing objects from selected regions and restoring those areas with content that seamlessly matches the surrounding patterns. As illustrated in Figure 4 of our submission, LaMa demonstrates outstanding performance on the Masking task, producing outputs that are nearly identical to the ground-truth masking results.
>
> However, LaMa's input is limited to masks alone, and it does not incorporate editing instructions. This limitation prevents it from adding new content or performing edits specified in the instructions for the Inpainting task. As a result, LaMa’s performance on the Inpainting task often appears as if it is merely copying the input document image, particularly when no meaningful instruction-driven modifications are made.
>
> In contrast, other baseline models, such as InpaintAnything, InstructPix2Pix, and ZONE, which are built upon Stable Diffusion, struggle with the complexity of document editing instructions. These instructions are typically longer and more intricate compared to those encountered during their training. Consequently, these models may distort the entire canvas or perform incorrect edits in the wrong regions, leading to results that deviate significantly from the intended outcome.

---

> ### Author Response · Authors · 2024-11-27
> **Author Response to Reviewer oGDA (Part 2)**
>
> > **Error Analysis & Insights to Optimize Model Performance**
>
> We sincerely thank the reviewer for their valuable suggestion. We are pleased to include a more detailed discussion on the error analysis and provide additional insights to optimize model performance. The discussion is presented below, and we have also incorporated this analysis into the revised submission (please refer to page 8~9).
>
> We’d like to note here that we observed that Stable Diffusion-based models struggle greatly with text rendering, particularly in the context of document editing. While prior works such as TextDiffuser [1,2] have explored text rendering using Stable Diffusion, these efforts primarily focus on short text snippets -- typically only two to three words -- and are exclusively trained on English text.
>
> In contrast, document editing tasks in our scenario often involve inpainting text elements that span entire paragraphs, posing a much greater challenge. Moreover, our ADOPD-Instruct dataset includes annotations for languages beyond English, incorporating non-alphabetic characters such as Korean, Japanese, and Chinese. These multilingual and multi-character requirements significantly increase the complexity of the editing instructions, exposing the limitations of existing image editing models.
>
> Notably, all document editing inferences in our experiments were conducted in a zero-shot setting, without any finetuning of the tested models. The observed suboptimal performance highlights the domain gap between the training data of current image editing models and the specific challenges of document editing tasks. This performance disparity can largely be attributed to the lack of annotated datasets tailored to the document domain, which restricts the ability of these models to generalize effectively.
>
> Our empirical analysis underscores the limitations of current image editing models in handling complex scenarios like visually-rich documents. To address this gap, we introduced the ADOPD-Instruct dataset, which we believe will serve as a valuable resource for advancing future models in this domain. By enabling more robust training and evaluation on document-specific tasks, ADOPD-Instruct has the potential to significantly improve the capabilities of image editing models in real-world applications.
>
> [1] TextDiffuser: Diffusion Models as Text Painters, NeurIPS 2023
>
> [2] TextDiffuser-2: Unleashing the Power of Language Models for Text Rendering, ECCV 2024

---

> ### Comment · Reviewer_oGDA · 2024-12-03
>
> Thanks for the authors' hard work. The rebuttal addressed some of my concerns, but the experiments in the paper are not sufficient. I choose to maintain the score.

---

### Official Review · Reviewer_8SC2 · 2024-11-01

**Soundness:** 3
**Presentation:** 3
**Contribution:** 2
**Rating:** 3
**Confidence:** 4

**Summary:**

The paper introduces a multimodal dataset called ADOPD-Instruct, designed to benchmark model capabilities for document editing. This dataset is derived from the existing ADOPD dataset (ICLR 2024) but has been re-annotated. The annotation process involves both an automated procedure using GPT-4o and human verification. Consequently, the dataset assesses the capabilities of MLLMs in instruction generation (referred to as "masking" in the paper) and the capabilities of editing models in instruction following (referred to as "inpainting" in the paper). Both text and non-text elements are considered. The study examines 8 MLLMs and 4 image editing models.

**Strengths:**

+ The task of document editing holds significant practical value in real-world applications, yet it remains under-explored.
+ Observing the limitations of current models in executing document editing tasks provides valuable insights for further research.
+ The paper extensively evaluates 8 MLLMs and 4 image editing models.

**Weaknesses:**

- Major: The reviewer recognizes the empirical contributions of this work. However, the paper falls short in terms of technical contributions and providing insightful knowledge to the community. Specifically, the dataset introduced in the paper is derived from an existing public dataset. While new annotations are provided, the dataset is relatively small in scale and offers limited insights for future research. Additionally, the evaluation of existing models highlights the challenges they currently face, but it does not offer any guidance on how the community could enhance model performance for these tasks.

- Although the work might be new in the field of visually-rich document editing, it is related to tasks such as text-rich and layout multimodal reasoning and editing. It is recommended to carefully discuss these previous works.

> TRINS: Towards Multimodal Language Models that Can Read. CVPR 2024.

> LayoutLLM: Layout Instruction Tuning with Large Language Models for Document Understanding. CVPR 2024.

**Questions:**

Please see the Weaknesses.

---

> ### Author Response · Authors · 2024-11-28
> **Author Response to Reviewer 8SC2 (Part 1)**
>
> > **Technical Contribution of Our Work**
>
> We respectfully disagree with the assertion that our work lacks significant technical contributions. The motivation for introducing a well-annotated document dataset is grounded in addressing the evident gap in studying document-related topics, as highlighted by our empirical results. The two tasks presented in our study are deliberate extensions of classic computer vision tasks, specifically adapted to the document domain:
> * Instruction Generation for Document Editing extends the traditional image comparison and differencing task into the document domain, incorporating the unique challenges associated with document editing.
> * Instruction-Following Document Editing builds upon the instruction-following image editing task, adapting it for the document domain with its intricate visual structures and multilingual text elements.
>
> The poor zero-shot inference performance of state-of-the-art models on these tasks underscores the significant domain gap between existing multimodal language models' training data and real-world document data. This observation emphasizes the importance of introducing datasets like ADOPD-Instruct to foster progress in this domain.
>
> The uniqueness of our ADOPD-Instruct dataset is multi-faceted:
> * **Entity-Level Focus**: ADOPD-Instruct is centered on entity-level document editing, providing fine-grained instructions that specify the location of the edit, the content of the edit, and a detailed step-by-step description of the editing process.
> * **Comprehensive Tasks**: It includes two common document editing subtasks—Masking and Inpainting—addressing edits for both text elements and non-text design elements.
> * **Multilingual Support**: While the instructions are primarily written in English, the dataset includes multilingual descriptions of the text elements being edited. This includes non-alphabetic characters such as Korean, Japanese, and Chinese, adding a layer of complexity that is critical for real-world applications.
>
> We believe that the combination of these features makes ADOPD-Instruct a valuable and unique resource for advancing document AI. Its contributions lie not only in its empirical findings but also in addressing foundational challenges in the field, enabling researchers to explore and develop models that better handle the complexities of document editing tasks. Thank you for the opportunity to elaborate on these aspects.

---

> ### Author Response · Authors · 2024-11-28
> **Author Response to Reviewer 8SC2 (Part 2)**
>
> > **The Scale of ADOPD-Instruct Dataset**
>
> Thank the reviewer for the comment. While we appreciate the reviewer’s perspective, we respectfully disagree with the assertion that the ADOPD-Instruct dataset is small in scale and offers limited insights for future research.
>
> As demonstrated in Table 1 of our submission, the ADOPD-Instruct dataset is the largest document datasets that include both entity-level annotations and well-annotated instructions. With 181k data items, it ranks third in size among the compared datasets and is of the same order of magnitude as the two larger datasets.
>
> Moreover, the combination of entity-level annotations and high-quality, detailed instructions provides unique value for research in document editing and related tasks. This distinctive characteristic not only supports the development of robust models for instruction-following document editing but also lays the groundwork for further exploration into complex document-related challenges.
>
> We believe the ADOPD-Instruct dataset offers significant potential to drive advancements in document AI and encourage innovative research directions in this domain. Thank you for the opportunity to clarify the contributions and scale of our dataset.
>
> > **Insights for Future Improvements**
>
> We sincerely thank the reviewer for the suggestion of including more discussion on insights for possible future improvements along this line. We are pleased to include a more detailed discussion on the error analysis and provide additional insights to optimize model performance. The discussion is presented below, and we have also incorporated this analysis into the revised submission (please refer to page 8~9).
>
> We’d like to note here that we observed that Stable Diffusion-based models struggle greatly with text rendering, particularly in the context of document editing. While prior works such as TextDiffuser [1,2] have explored text rendering using Stable Diffusion, these efforts primarily focus on short text snippets -- typically only two to three words -- and are exclusively trained on English text.
> In contrast, document editing tasks in our scenario often involve inpainting text elements that span entire paragraphs, posing a much greater challenge. Moreover, our ADOPD-Instruct dataset includes annotations for languages beyond English, incorporating non-alphabetic characters such as Korean, Japanese, and Chinese. These multilingual and multi-character requirements significantly increase the complexity of the editing instructions, exposing the limitations of existing image editing models.
>
> Notably, all document editing inferences in our experiments were conducted in a zero-shot setting, without any finetuning of the tested models. The observed suboptimal performance highlights the domain gap between the training data of current image editing models and the specific challenges of document editing tasks. This performance disparity can largely be attributed to the lack of annotated datasets tailored to the document domain, which restricts the ability of these models to generalize effectively.
>
> Our empirical analysis underscores the limitations of current image editing models in handling complex scenarios like visually-rich documents. To address this gap, we introduced the ADOPD-Instruct dataset, which we believe will serve as a valuable resource for advancing future models in this domain. By enabling more robust training and evaluation on document-specific tasks, ADOPD-Instruct has the potential to significantly improve the capabilities of image editing models in real-world applications.
>
> [1] TextDiffuser: Diffusion Models as Text Painters, NeurIPS 2023
>
> [2] TextDiffuser-2: Unleashing the Power of Language Models for Text Rendering, ECCV 2024
>
>
>
> > **Discuss More Related Work**
>
> We sincerely thank the reviewer for their valuable suggestion. In response, we have incorporated a discussion of these two works into the Related Work section of the revised submission (please see page 2, lines 121–124).

---

### Official Review · Reviewer_tkNm · 2024-11-03

**Soundness:** 3
**Presentation:** 1
**Contribution:** 3
**Rating:** 5
**Confidence:** 4

**Summary:**

This paper presents ADOPD-Instruct, a comprehensive large-scale multimodal dataset aimed at visually-rich document editing tasks based on the instruction. The dataset stands out for its detailed annotations, step-by-step edit instructions, and its ability for this rich-text image editing tasks.
The authors also evaluate the performance of existing MLLMs on the editing instruction generated tasks and image editing models on document editing tasks, highlighting key limitations in current models and underscoring the dataset’s importance for advancing this field.

**Strengths:**

ADOPD-Instruct provides a novel contribution that fills a critical gap in multimodal document editing by addressing the scarcity of datasets that support entity-level text edits in document images.

The dataset presented in the paper shows to be highly applicable for real-world scenarios, making it particularly relevant to document understanding and editing fields.

The experimental section offers valuable insights on current model limitations, revealing significant challenges in current methods, particularly around instruction-following and complex text rendering.

**Weaknesses:**

- Lack of Clarity in Data Generation:  The dataset generation process is unclear, particularly regarding instruction and image generation. A thorough description of the data curation process would improve the clarity of the paper. Given the paper's length (8 pages), additional information on dataset curation could be great. A more detailed explanation in the revised version would positively influence my review score.

- Image Sources and Annotation Details: More details of the image sources and human annotators to imporve the soundness of the paper.

- The paper could benefit from a more detailed classification of edit tasks beyond just Masking and Inpainting for single text and non-text elements.

- More related works on image editing model and datasets[1-4], as well as the text rendering ability of generative models[5-8] will benfit to the paper.

[1]MagicBrush: A Manually Annotated Dataset for Instruction-Guided Image Editing

[2]Emu Edit: Precise Image Editing via Recognition and Generation Tasks

[3]HQ-Edit: A High-Quality Dataset for Instruction-based Image Editing

[4]UltraEdit: Instruction-based Fine-Grained Image Editing at Scale

[5]TextDiffuser-2: Unleashing the Power of Language Models for Text Rendering

[6]GlyphDraw2: Automatic Generation of Complex Glyph Posters with Diffusion Models and Large Language Models

[7]A spark of vision-language intelligence: 2-dimensional autoregressive transformer for efficient finegrained image generation

[8]GPT4O

**Questions:**

1. How were the source and target images generated, especially given the high quality of examples in the paper? The data generation process remains ambiguous after reading.

2. Use of GPT-4o Instructions: Given the failure rate of GPT-4o instructions (nearly 60%, per Table 2), what is the justification for relying on these instructions, especially when human annotators are required to review and refine them? I also wonder the performance of gpt4o on ADOPD-Instruct for image editing.

3. Do the Masking and Inpainting tasks share the same image sources, with the tasks simply reversed?

4. Since ADOPD-Instruct focuses on Masking and Inpainting tasks with changes only occurring in the masked areas, are the metrics in Table 5 appropriate for evaluation? Local metrics such as SSIM or CLIP score in the masked area might provide a more accurate assessment. The best performance of LaMa may be due to simply copying the input image, resulting in relatively low scores.

5. Can more details be provided about the specific split of ADOPD-Instruct used in the evaluation?

---

> ### Author Response · Authors · 2024-11-27
> **Author Response to Reviewer tkNm (Part 1)**
>
> > **More Details Regarding Dataset Curation Process**
>
> We thank the reviewer for raising this important point. To provide greater clarity, we have included a screenshot of the human curation interface in Figure 5 (page 17 of the submission).
>
> During the human curation process, annotators are presented with the document images both before and after the edits, along with the corresponding GPT-4o-generated instruction awaiting refinement. To assist annotators in clearly identifying the edited elements, we apply a translucent mask over the document image, highlighting the region where the edit has occurred. For example, in the case illustrated in Figure 5, the edit region is located in the top-right corner, just below "999".
>
> The annotators review the highlighted area and the accompanying GPT-4-O-generated instruction. They then refine the instruction by correcting inaccuracies and documenting any detected errors. This structured annotation process ensures that the resulting instructions are accurate and align with the edits made to the document.
>
> We believe that this combination of automated instruction generation and human refinement enhances both efficiency and quality, contributing to the value of the ADOPD-Instruct dataset. Thank you for giving us the opportunity to elaborate on this process.
>
>
> > **Document Image Source**
>
> The source and target document images, along with the segmentation masks, are provided by the ADOPD dataset [1]. These images are originally sourced from the LAION-HR dataset [2], and their entity-level segmentation annotations were meticulously created by human annotators.
>
> The unique contribution of our ADOPD-Instruct dataset lies in its extension of the ADOPD dataset to address document editing tasks. Specifically, we have designed two novel document editing tasks and provided detailed, step-by-step annotations that describe the entity-level editing processes. These annotations encompass both text elements and non-text design elements, enabling a deeper exploration of document editing challenges.
>
> We believe this setup offers a valuable resource for advancing research in document editing, and we are grateful for the opportunity to clarify this aspect of our dataset.
>
> [1] ADOPD: A Large-Scale Document Page Decomposition Dataset. ICLR 2024.
>
> [2] https://huggingface.co/datasets/laion/laion-high-resolution.
>
>
>
>
> > **More Detailed Classification of Editing Task**
>
> Thank you for your suggestion! In future work, we could provide more detailed classification by the type of the design elements being edited (e.g., title, subtitle, paragraph, table, sidebar, footnote, background, decoration, separator, etc.)
>
> > **More Related Work**
>
> We sincerely thank the reviewer for the suggestion on related works. Following the reviewer’s suggestion, we have incorporated a discussion of these two works into the Related Work section of the revised submission (see page 2, lines 143-153).
>
> > **Motivation for Starting from GPT-4o Instructions**
>
> Thank you for your insightful comment. We acknowledge the limitations of GPT-4o instructions and appreciate the opportunity to clarify our rationale for leveraging them in this work. We choose to use GPT-4o to draft the document editing instructions mainly for the following two reasons:
>
> (1) *Efficiency in Instruction Drafting*: While GPT-4o has a failure rate of nearly 60% as noted in Table 2, it still provides a relatively complete structure for editing instructions. Based on feedback from human annotators, editing and refining these drafts is faster and easier than writing instructions entirely from scratch. This workflow significantly reduces the annotators’ cognitive load and expedites the annotation process, making GPT-4o a practical starting point despite its imperfections.
>
>
> (2) *Support for Multilingual Content*: Although the instructions are primarily written in English, the document editing tasks often involve content in multiple languages. This is particularly true for edits on text elements in our ADOPD-Instruct dataset, which includes multilingual documents with non-alphabetic characters such as Korean, Japanese, and Chinese. Identifying and transcribing these characters accurately can be challenging for human annotators. However, GPT-4o demonstrates strong OCR capabilities, enabling it to detect and transcribe foreign characters into the initial instruction drafts. This preliminary transcription facilitates the human curation process, especially in multilingual contexts. (We have included this in the revised draft, please refer to page 4.)
>
>
> By utilizing GPT-4o as a foundational tool, we aim to balance efficiency and accuracy in generating high-quality document editing instructions. We appreciate the reviewer's suggestion and hope this explanation highlights the practical contributions of GPT-4o in our workflow.
>
>
> > **Image Source for the Masking and Inpainting Tasks**
>
> Yes, the Masking and Inpainting tasks share the same document image source.

---

> ### Author Response · Authors · 2024-11-27
> **Author Response to Reviewer tkNm (Part 2)**
>
> > **Use SSIM and CLIP score to Evaluate Image Similarity in Table 5**
>
> Thank you for the thoughtful suggestion regarding evaluation metrics! We appreciate the reviewer’s insight and are pleased to incorporate additional metrics for a more comprehensive assessment.
>
> We would like to highlight that Table 5 in the original submission already includes the CLIP image similarity score as part of our evaluation. In response to the reviewer's recommendation, we have now computed and reported the SSIM scores as well. The SSIM results are consistent with the trends observed in other metrics such as FID, LPIPS, PSNR, and CLIP image similarity. Specifically, LaMa demonstrates the best performance, followed by InpaintAnything.
>
> | Task       | Model           | Text Elements | Non-Text Elements |
> |------------|-----------------|---------------|-------------------|
> | Masking    | LaMa            | **99.75**     | **99.96**         |
> |            | InpaintAnything | *96.73* | *95.75*      |
> |            | InstructPix2Pix | 74.47         | 72.84             |
> |            | ZONE            | 87.49         | 86.33             |
> | Inpainting | LaMa            | **97.23**     | **95.61**         |
> |            | InpaintAnything |*95.16* |*91.67*      |
> |            | InstructPix2Pix | 72.98         | 71.99             |
> |            | ZONE            | 86.40         | 86.06             |
>
> We have also incorporated this in the revised draft. We believe these additional results further validate the robustness of our evaluation and thank the reviewer for encouraging us to strengthen our analysis.
>
>
>
> > **Why does LaMa have the best performance?**
>
> Thank you for raising this insightful concern! We would like to elaborate on why LaMa achieves the best performance in the reported metrics. We have also added the following discussion to the revised submission (please refer to page 9).
>
> LaMa is specifically designed and trained for mask inpainting tasks, excelling at removing objects from selected regions and restoring those areas with content that seamlessly matches the surrounding patterns. As illustrated in Figure 4 of our submission, LaMa demonstrates outstanding performance on the Masking task, producing outputs that are nearly identical to the ground-truth masking results.
>
> However, LaMa's input is limited to masks alone, and it does not incorporate editing instructions. This limitation prevents it from adding new content or performing edits specified in the instructions for the Inpainting task. As a result, LaMa’s performance on the Inpainting task often appears as if it is merely copying the input document image, particularly when no meaningful instruction-driven modifications are made.
>
> In contrast, other baseline models, such as InpaintAnything, InstructPix2Pix, and ZONE, which are built upon Stable Diffusion, struggle with the complexity of document editing instructions. These instructions are typically longer and more intricate compared to those encountered during their training. Consequently, these models may distort the entire canvas or perform incorrect edits in the wrong regions, leading to results that deviate significantly from the intended outcome.
>
> We hope this explanation clarifies the observed performance trends and emphasizes the distinct strengths and limitations of the models under evaluation. Thank you for highlighting this important aspect!
>
> > **ADOPD-Instruct Evaluation Split**
>
> In our submission, the evaluation split for the empirical study of current models' performance on document editing tasks was randomly sampled from the ADOPD-Instruct dataset. Specifically, we selected 4,000 data samples for evaluation, ensuring a balanced distribution across the two subtasks and the two types of design elements:
>
> | Document Editing Task | Text Elements | Non-Text Elements |
> |-----------------------|---------------|-------------------|
> | Masking               | 1k            | 1k                |
> | Inpainting            | 1k            | 1k                |

---

### Official Review · Reviewer_YWzh · 2024-11-07

**Soundness:** 2
**Presentation:** 3
**Contribution:** 2
**Rating:** 5
**Confidence:** 3

**Summary:**

The paper introduces ADOPD-Instruct, a novel large-scale multimodal dataset designed to address the complexities of visually-rich document editing. The authors have curated a collection of visually-rich documents, precise masks highlighting elements to be edited, and human-curated instructions that target both text and non-text design elements. The paper also presents an extensive evaluation of current Multimodal Large Language Models and image editing models, revealing the challenges these models face in accurately generating and following detailed instructions for document editing.

**Strengths:**

1. ADOPD-Instruct is quite useful in downstream tasks requiring document-level image editing. It stands out for its scale and the level of detail in its annotations. The inclusion of both text and non-text elements, along with segmentation masks and human-curated instructions, makes it a rich resource for researchers in document editing.
2. The annotation process is clear and easy to understand.
3. The paper benchmarks a wide range of available models, showing that the proposed task is challanging according to the results.

**Weaknesses:**

- There is a lack of clarity regarding the involvement of human annotators in refining the instructions generated by GPT-4-O, as indicated in Table 2, which shows that a significant portion (>60%) of the instructions produced by the GPT model are incorrect.
- The need for an instruction generation task is also difficult to comprehend, particularly in the context of document editing. Typically, initial instructions are crafted by humans, so if the edited image and mask are already prepared, it raises the question of why further editing is necessary. This task setup appears to be rather disconnected from real-world applications.

**Questions:**

1. Why separate the image editing task into detailed instruction generation and instruction-following document editing?

---

> ### Author Response · Authors · 2024-11-27
> **Author Response to Reviewer YWzh (Part 1)**
>
> > **Details Regarding Human Annotation**
>
> We thank the reviewer for raising this important point. To provide greater clarity, we have included a screenshot of the human curation interface in Figure 5 (page 17 of the submission).
>
> During the human curation process, annotators are presented with the document images both before and after the edits, along with the corresponding GPT-4o-generated instruction awaiting refinement. To assist annotators in clearly identifying the edited elements, we apply a translucent mask over the document image, highlighting the region where the edit has occurred. For example, in the case illustrated in Figure 5, the edit region is located in the top-right corner, just below "999".
>
> The annotators review the highlighted area and the accompanying GPT-4-O-generated instruction. They then refine the instruction by correcting inaccuracies and documenting any detected errors. This structured annotation process ensures that the resulting instructions are accurate and align with the edits made to the document.
>
> We believe that this combination of automated instruction generation and human refinement enhances both efficiency and quality, contributing to the value of the ADOPD-Instruct dataset. Thank you for giving us the opportunity to elaborate on this process.
>
>
> > **Motivation of Designing the Document Editing Instruction Generation Task**
>
> We appreciate the reviewer’s feedback and would like to clarify the motivation behind this task.
>
> The document editing instruction generation task draws inspiration from the computer vision task of image comparison and differencing, where the goal is to describe differences between two images in natural language. We adapt this concept to the domain of documents, where differences can encompass intricate visual elements, complex layouts, or modifications to text boxes.
>
> This task addresses a critical gap: enabling scalable, automatic annotation for unseen documents. In real-world scenarios, users often provide documents without pre-existing annotations. By generating detailed, natural language descriptions of editing instructions, our task aims to support the development of open-source models capable of producing high-quality annotations.
>
> Our experiments demonstrate that current open-source large multimodal models fall short in this area (Table 4), and closed-source models, such as OpenAI's offerings, frequently make errors. Moreover, relying on proprietary services and manual curation is costly and limits scalability. By advancing open-source solutions, this task facilitates a more efficient and cost-effective approach to generating detailed and accurate descriptions, ultimately enabling broader adoption and application in real-world document editing scenarios.
>
> We hope this clarifies the practical relevance and the significant contribution of this task to the field. Thank you for raising this important point.

---

> ### Author Response · Authors · 2024-11-27
> **Author Response to Reviewer YWzh (Part 2)**
>
> > **Motivation of Separating the Document Editing Process into Two Subtasks**
>
> We sincerely thank the reviewer for their insightful question. Our primary motivation for separating the document editing process into two subtasks is rooted in the unique challenges and underexplored nature of the document domain. This separation allows for a comprehensive investigation of document-specific tasks, from dataset creation and task design to model training, paving the way for advancements in this area.
>
>  The first subtask of “Instruction Generation for Document Editing” extends the traditional computer vision task of image comparison and differencing into the document domain. As shown in Table 4, current large multimodal language models perform poorly on this task, highlighting a significant domain gap between their training data and real-world scenarios such as the document domain represented by our ADOPD-Instruct dataset. Addressing this gap is critical for improving the capabilities of these models in practical applications.
>
>
> The second subtask of “Instruction-Following Document Editing” builds on the traditional instruction-following image editing task, adapted for the document domain. As demonstrated in Table 5 and Figure 4, existing image editing models face substantial challenges when tackling document editing tasks. These difficulties are particularly evident when models are required to insert long text snippets in multilingual contexts, involving complex visual layouts and non-alphabetic characters (e.g., Korean, Japanese, and Chinese). Such scenarios present significant hurdles for current models, underscoring the importance of our dataset in addressing these limitations.
>
> By decoupling the document editing process into these two subtasks, we aim to systematically address the unique challenges of the document domain and highlight critical gaps in existing models. This setup emphasizes the importance of our ADOPD-Instruct dataset in driving future advancements in document AI research. Thank you for the opportunity to elaborate on this important design choice.

---

### Meta-Review · Area_Chair_nrdS · 2024-12-17

**Metareview:**

This paper introduces a dataset for document editing and establishes evaluations of current Multimodal Large Language Models (MLLMs) and image editing models to assess their performance on this task. While the reviewers acknowledge the value of the dataset, they highlight several weaknesses: (1) a lack of significant technical contribution, (2) insufficient insights on improving task performance, and (3) limited empirical verification. Two reviewers find that the paper is below ICLR standards.

**Additional Comments On Reviewer Discussion:**

The author submitted the rebuttal not in time, resulting in minimal feedback from the reviewers. Both reviewers responded, stating that their main concerns were not addressed.

---

### Decision · Program_Chairs · 2025-01-22

Reject